# The Natural HASPIN Inhibitor Coumestrol Suppresses Intestinal Polyp Development, Cachexia, and Hypogonadism in a Mouse Model of Familial Adenomatous Polyposis (*Apc^Min/+^*)

**DOI:** 10.3390/biology13090736

**Published:** 2024-09-20

**Authors:** Hiromitsu Tanaka, Shunsuke Matsuyama, Tomoe Ohta, Keisuke Kakazu, Kazutoshi Fujita, Shinichiro Fukuhara, Tetsuji Soda, Yasushi Miyagawa, Akira Tsujimura

**Affiliations:** 1Faculty of Pharmaceutical Sciences, Nagasaki International University, Sasebo 859-3298, Nagasaki, Japan; matsuyama.s.9808@gmail.com (S.M.); ohta-t@mb.kyoto-phu.ac.jp (T.O.); k.kakazu@niu.ac.jp (K.K.); 2Department of Urology, Kindai University Faculty of Medicine, Osakasayama 589-8511, Osaka, Japan; kfujita@med.kindai.ac.jp; 3Department of Urology, Osaka University Graduate School of Medicine, Suita 565-0871, Osaka, Japan; fukuhara@uro.med.osaka-u.ac.jp (S.F.); miyagawa@uro.med.osaka-u.ac.jp (Y.M.); 4Department of Urology, Osaka Central Hospital, Kita-ku, Osaka 530-0001, Japan; tetsujisoda@gmail.com; 5Department of Urology, Sumitomo Hospital, Kita-ku, Osaka 530-0005, Japan; 6Department of Urology, Juntendo University Urayasu Hospital, Chiba 279-0021, Japan

**Keywords:** androgen, histone H3, anti-aging, cancer, protein kinase, testis

## Abstract

**Simple Summary:**

Approximately half of the populations of developed countries contract cancer, with a very high proportion of colorectal cancer among all cancer types. Food choices must be improved to maintain good health. HASPIN inhibitors suppress the proliferation of various cancer cells. In this study, the antitumor effect of ingesting bean sprouts containing the HASPIN inhibitor coumestrol was investigated using a mouse model of familial adenomatous polyposis (*Apc^Min/+^*). The results indicated that ingesting a diet including bean sprouts suppressed the development of intestinal polyps, cachexia, and hypogonadism in mice. These findings demonstrated that bean sprouts are a beneficial food for preventing cancer and are expected to be applicable in humans.

**Abstract:**

(1) Background: HASPIN kinase is involved in regulating spindle function and chromosome segregation, as well as phosphorylating histone H3 at Thr3 in mitotic cells. Several HASPIN inhibitors suppress cancer cell proliferation. It was recently reported that coumestrol from bean sprouts inhibits HASPIN, and a cultivation method for bean sprouts containing large amounts of coumestrol has been established. Here, we showed the effects of bean sprout ingestion on intestinal polyp development, cachexia, and hypogonadism in a mouse model of familial adenomatous polyposis (*Apc^Min/+^*). (2) Methods: *Apc^Min/+^* mice were randomized into control and treatment groups. Mice in the control group were given the standard diet, while those in the treatment group were given the same standard diet with the addition of 15% bean sprouts. Treatments were commenced at 7 weeks old and analyses were performed at 12 weeks old. (3) Results: ingesting bean sprouts suppressed the development of intestinal polyps, cachexia, and hypogonadism, and also increased serum levels of testosterone in male wild-type and *Apc^Min/+^* mice. (4) Conclusions: ingesting bean sprouts helps prevent cancer and increases serum levels of testosterone in a mouse model. These results are expected to be applicable to humans.

## 1. Introduction

The medicinal properties of bean sprouts were described in Japanese literature 1500 years ago. After World War II, bean sprouts were eaten as a nutritious and readily available vegetable, even during food shortages in Japan. Currently, bean sprouts are used in various dishes for their nutritional value. Bean sprouts contain polyphenols [1], and we have focused our research on the HASPIN inhibitor coumestrol [2]. It has been reported that coumestrol influences steroid synthesis due to its structural similarity with estrogens [3,4], as well as its ability to bind estrogen receptors and promote bone formation [5]. In addition, coumestrol inhibits cholinesterase [6] and activates SIRT1 [7]. Coumestrol inhibits HASPIN serine-threonine kinase, which is involved in forming the chromosomal passenger complex by the phosphorylation of histone H3 [8]. HASPIN inhibitors prevent the growth of various cancers [9]. HASPIN is expressed in the hippocampus and phosphorylates the tau protein [10]. Ingestion of bean sprouts containing coumestrol suppresses the development of spatial–cognitive dysfunction in the 5xFAD Alzheimer’s disease mouse model [10].

HASPIN phosphorylates histone H3 at Thr3 in mitotic cells. While the level of H3-T3 phosphorylation peaks during mitosis, it also occurs during interphase [11] and G0 [12], and even non-cycling tissues and the function of their primary cilia are affected [12]. HASPIN plays a role in the Thr127 phosphorylation of germ cell-specific TH2A [13] and chromosome segregation during meiosis [14]. It is also involved in the phosphorylation of hyaluronan-binding protein 1 (HABP1/p32/gC1qR/C1QBP), which plays an important role in fertilization [15]. It plays multiple roles as a serine/threonine kinase in spermatogenesis; it is widely conserved in animals, yeasts, and plants [16], and plays various regulatory roles in cellular function. It also has functions that are relevant to carcinogenesis, including CPC recruitment [17,18,19], tissue morphogenesis and homeostasis [20,21], Pds5 and cohesin regulation [22,23], and polycomb regulation [11]. However, no critical defects have been observed in HASPIN gene-disrupted mice, indicating that the functions of HASPIN are complemented by other molecules during normal cellular activity [24].

We recently established a method for the cultivation of bean sprouts containing high levels of coumestrol, enabling its more efficient ingestion. Coumestrol inhibits HASPIN kinase activity and is speculated to have beneficial health effects. Intraperitoneal administration of the artificial HASPIN inhibitor CHR-6494 was shown to suppress colon cancer in a mouse model of familial adenomatous polyposis (*Apc^Min/+^)* [25]. The *Apc^Min/+^* mouse is a familial colon cancer mouse model with a nonsense mutation at codon 850 in the *Apc* gene [26]. APC protein forms a complex with various other proteins and is involved in the control of signal transduction pathways [26,27]. It may be a tumor suppressor involved in the earliest stages of human colorectal cancer development. Here, the cancer-suppressing effects of mung bean (*Vigna radiata*) sprouts cultivated to contain high levels of coumestrol were investigated in the *Apc^Min/+^* (C57BL/6J) mouse model of human colorectal cancer.

## 2. Materials and Methods

### 2.1. Animals

*Apc^Min/+^* mice were obtained from Jackson Laboratories (Bar Harbor, ME, USA) and used for experiments after they had been bred with C57BL/6J mice in our animal facility. The C57BL/6J mice were purchased from Japan SLC, Inc. (Hamamatsu, Japan). The mice were sacrificed by cervical dislocation immediately before the experiments. All animal experiments conformed to the Guide for the Care and Use of Laboratory Animals and were approved by the Institutional Committee of Laboratory Animal Experimentation and Research Ethics Committee of Nagasaki International University (no. 148). This study did not include any human experiments performed by any of the authors. The mice were maintained under specific pathogen-free conditions in the animal experimentation facility of Nagasaki International University, with temperature and lighting controlled during the experiment. The mice were provided with food and water ad libitum. Genotypes were determined by the polymerase chain reaction (PCR) using the conditions reported previously [28]. The mice were fed the specialized laboratory rodent diet 5L37 LABDIET (Japan SLC, Inc.), which is available internationally as a LabDiet^®^ product (Land O’ Lakes, Inc., Arden Hills, MN, USA). The solid diet was mixed with 15% sprout powder and 10% potato starch as a binding agent in 5L37 LABDIET, and treated at 80 °C for 10 h. The mice were fed bean-containing diets beginning at 7 weeks after birth, and biological analyses were performed at 12 weeks.

### 2.2. Cultivation of the Bean Sprouts

Bean sprouts were purchased from the Kawano Shop Co., Ltd. (Nagasaki, Japan) to grow mung beans (*V. radiata*); they were cultivated for an additional 3 days after purchase to ensure they contained large amounts of coumestrol (approximately 200 μg/g on a dry weight basis). Sprout cultivation and coumestrol measurement were performed as reported previously [29]. The bean sprouts were cultivated under natural light in a temperature-controlled room in trays containing tap water to a depth of approximately 5 mm. The cultivated bean sprouts were dried at 80 °C for 12 h and then ground in a mill.

### 2.3. Intestinal Polyp Counts

Intestinal polyps were counted using the published method [30]. The length of freshly removed intestine was measured, and each small intestine was divided into five equal segments. The segments were incised longitudinally, washed with phosphate-buffered saline, laid flat on filter paper, and fixed for 24 h with 10% neutral-buffered formalin. The fixed intestinal segments were stained with 1% methylene blue and examined for tumors by gross inspection and light microscopy. Polyps more than 2 mm in diameter were classified as precancerous colon polyps [31].

### 2.4. Histological Observations of the Testes and Epididymides

The testes were collected from the scrotum, and the weights of the testes, epididymides, and epididymal adipose tissue were measured. Bouin’s solution-fixed mouse testes were cut into sections 8 μm thick and mounted on silane-coated slides. The slides were stained with hematoxylin and eosin, and then examined under a microscope.

### 2.5. Serum Testosterone Measurements

Serum testosterone levels were measured by liquid chromatography–tandem mass spectrometry (Oriental Yeast Co., Ltd., Tokyo, Japan).

### 2.6. Western Blotting

Protein samples obtained from the organs of adult mice were sonicated on ice in a TBS buffer (10 mM Tris-HCl, pH 7.5). After centrifugation, the protein concentration of each supernatant was estimated by the Bradford Protein Assay (Nacalai Tesque Inc., Kyoto, Japan). All extracts (50 μg protein) were subjected to sodium dodecyl sulfate–polyacrylamide gel electrophoresis and the proteins were transferred onto membranes. The membranes were blocked with blocking solution (Nacalai Tesque) and reacted with dilute anti-HASPIN (#52601; Cell Signaling Technology, Danvers, MA, USA) or anti-actin antibody (sc-1616; Santa Cruz Biotechnology, Dallas, TX, USA) (1:1000) in Can Get Signal (ToYoBo, Osaka, Japan) overnight at 4 °C. After washing in TBS containing 0.2% Tween-20, the membranes were incubated with peroxidase-conjugated anti-rat antibody (Dako, Tokyo, Japan) or goat (Dako) immunoglobulin (1:1000) for 1 h at room temperature in Can Get Signal. After further washing, the antigen–antibody complexes were detected using ECL Select (Cytiva Amersham, Marlborough, MA, USA). The signals were observed using Image Quant LAS4000 (Fujifilm, Tokyo, Japan).

### 2.7. Statistical Analyses

Data are expressed as the mean ± standard deviation. Statistical analysis was performed using the Student’s *t*-test and one-way analysis of variance (ANOVA) with Dunnett’s or Tukey’s test. In all analyses, *p* < 0.05 was taken to indicate statistical significance.

## 3. Results

### 3.1. Body Weight

*Apc^Min/+^* mice showed normal growth to adulthood, similar to wild-type controls, but lost body weight and became infertile with the development of colon cancer [27] (Figure 1). We investigated the effects of ingestion of bean sprouts on the appearance of the mice. In the treatment group, mice were fed the diet with bean sprouts between 7 and 12 weeks of age. *Apc^Min/+^* mice developed cachexia during the period of treatment [27] (Figure 1 and Figure 2, Appendix A). We measured the body weight of mice at the end of the experiment. The results showed significant recovery of body weight in mice fed the diet with bean sprouts compared with the controls (Figure 2, and Appendix A). Both body weight and epididymal fat content were lower in wild-type and *Apc^Min/+^* mice that had a diet containing bean sprouts than in those given the standard diet (Table 1). Mice fed the diet containing bean sprouts continued to grow and exhibited minimal cachexia after 12 weeks (Figure 1 and Figure 2, and Appendix A).

### 3.2. Intestinal Polyps

Intestinal polyps were observed in *Apc^Min/+^* mice fed the diet with bean sprouts between 7 and 12 weeks of age. Intestinal polyps become larger with age in *Apc^Min/+^* mice [31]. We investigated the effects of bean sprout ingestion on the number of polyps more than 2 mm in diameter, defined as precancerous colon polyps [31] ( Appendix A). The numbers of intestinal polyps in the *Apc^Min/+^* mice are listed in Figure 3 and Appendix A. Significantly fewer polyps were observed in the small intestines of mice fed the diet with bean sprouts than in those given the control diet. The intestine was approximately 20% longer in mice fed the diet with bean sprouts than those given the standard diet (Figure 4, Appendix A).

### 3.3. Observations of Testes and Epididymides

*Apc^Min/+^* mice develop hypogonadism [32,33]. The HASPN inhibitor, CHR-6494, suppresses the development of hypogonadism [25]. We investigated the effects of bean sprout ingestion between 7 and 12 weeks of age on the progression of hypogonadism in *Apc^Min/+^* mice. The testicular and epididymal weights significantly recovered in mice fed the diet with bean sprouts compared to those fed the control diet (Table 1, Appendix A). Histochemical observations revealed that spermatogenesis recovered from hypogonadism in *Apc^Min/+^* mice fed the diet with bean sprouts (Figure 5). Serum level of testosterone were significantly increased in the mice fed the diet with bean sprouts compared to controls (Table 2).

### 3.4. HASPIN Expression

HASPIN is exclusively expressed in haploid germ cells and not in undifferentiated testicular germ cells (Figure 6) [34]. The expression levels of this protein are correlated with the proliferative state of the tissue [35]. HASPIN expression was not observed in the testes of *Apc^Min/+^* mice containing undifferentiated testicular germ cells but was observed in the intestine of *Apc^Min/+^* mice in which polyps had developed (Figure 6). Its expression in the testes and intestines of wild-type mice fed the diet containing bean sprouts was unchanged from that in mice fed the standard diet (Figure 6).

## 4. Discussion

The high incidence of cancer is related to a lack of biological adaptation to the long human lifespan. To achieve a human society with a longer lifespan that is stress-free, it is essential to reduce the incidence of cancer and maintain a healthy lifespan in older adults. Advances in cancer treatment technology have led to the development of drugs that inhibit cancer cell proliferation signals, radiotherapy targeting cancer cells, and drugs that prevent cancer from invading the immune system [36]. Various cancer prevention and treatment methods are in development, including chimeric antigen receptor T-cell therapy, which builds immunity against cells, and methods to detect the presence of small numbers of cancer cells in the body. However, the incidence of cancer continues to increase, and there is a need for the development of cheaper preventive and treatment methods with fewer side effects. It has recently been reported that HASPIN inhibitors specifically inhibit the proliferation of various cancer cells [9]. HASPIN is conserved throughout eukaryotes and has diverse functions [16]. As HASPIN binds several proteins in vitro and uses various proteins as phosphorylation substrates [15], it has been suggested that abnormal HASPIN expression can cause substantial cellular damage. However, no major phenotype was observed in HASPIN gene-disrupted mice [24]. These findings imply that the function of HASPIN is complemented by other proteins in normally differentiated cells. Alternatively, abnormal expression impaired cellular function. In this study, Western blotting of the small intestine with polyps revealed the expression of HASPIN, indicating that HASPIN may be a useful cancer marker.

Based on the results of these basic studies, HASPIN is considered a promising target molecule for anticancer drugs [9], and low-molecular compounds inhibiting HASPIN have been reported [37]. The natural polyphenol coumestrol was recently reported to inhibit HASPIN [2]. To examine the possibility of preventing cancer using natural substances, we examined coumestrol-rich plants, improved cultivation methods, and established methods for cultivating bean sprouts that contain high amounts of coumestrol [29]. Here, bean sprouts were orally administered to *Apc^Min/+^* mice, a model of familial colorectal cancer, and the cancer suppressive effect was analyzed. As *Apc^Min/+^* mice develop testicular atrophy at a relatively young age, we observed the effects of feeding coumestrol-rich bean sprouts on hypogonadism. The results show that the development of polyps, cancer-induced cachexia, and testicular atrophy were suppressed in *Apc^Min/+^* mice. The APC protein is a tumor suppressor that regulates the Wnt signaling pathway through the formation of complexes with various other proteins, and plays a role in the early stages of colorectal cancer development. Wnt/β-catenin signaling is also known to play an important role in the differentiation and proliferation of Leydig cells in the testis [26]. As administration of CHR-6494 suppressed the development of polyps and hypogonadism in *Apc^Min/+^* mice, HASPIN may not only function in chromosome segregation but also participate in the Wnt/β-catenin signaling pathway. Our results are identical to findings reported previously for the HASPIN inhibitor, CHR6494 [25]. Therefore, the HASPIN inhibitor coumestrol, within bean sprouts, is considered to have been the main factor involved in suppressing cancer.

Approximately 150 μg of CHR6494 was administered intraperitoneally each day (50 mg/kg = CHR6494/body weight), assuming a mouse body weight of 30 g. The HASPIN inhibitory effect (ID_50_ 50 μM) of coumestrol was sufficient to suppress cancer, although the ID_50_ was nearly 1000-fold lower than that of CHR6494 (approximately 50 nM) in suppressing the proliferation of cultured cancer cells [2,38]. The length of the intestine increased after oral administration of bean sprouts. Ingestion of bean sprouts resolved the cachexia and caused the *Apc^Min/+^* mice to gain weight. In contrast, the body weight of wild-type mice decreased after consumption of the bean sprouts. Coumestrol binds to estrogen receptors and has various physiological effects [1,29]. Our results may have been due to the effects of dietary fiber and the ingestion of polyphenols, as well as the multiple pharmacological effects of coumestrol as a HASPIN inhibitor. Moreover, the anticancer effects of bean sprout ingestion may have been due to interactions between coumestrol and other nutrients, and/or metabolites produced by intestinal bacteria.

Based on conversion of bean sprout powder in this mouse experiment to human body weight, a human would need to ingest approximately 1.5 kg of dried bean sprouts per day to achieve an equivalent dose. It would be difficult for humans to eat this much bean sprout powder. In the future, it will be necessary to examine the anticancer preventive effect in mice after oral administration of smaller amounts of bean sprout powder; human clinical trials regarding the anticancer preventive effects of consuming appropriate amounts of bean sprout powder will also be needed.

Although treatment methods, such as polypectomy, have been established for colorectal cancer, it is important to develop effective treatments that are tailored to different cancers. Therefore, the development of HASPIN inhibitors, such as coumestrol, will be useful in preventing colorectal cancer and in suppressing the risk of recurrence after surgical treatment. Although it has been reported that HASPIN inhibitors, including coumestrol, suppress the growth of prostate cancer and hormone-dependent breast cancer cells [39,40,41,42], the ingestion of bean sprouts increased serum testosterone levels in male mice. The effects of bean sprouts on these cancers must be analyzed carefully. Adolescent and young adult patients with advanced cancer often develop hypogonadism before treatment, and many of the anticancer drugs currently in use destroy reproductive cells. This destruction may be permanent, even after the cancer has been cured, leading to infertility. HASPIN inhibitors, such as coumestrol, may be useful as cancer treatment drugs that restore fertility in such patients. It would be interesting to analyze the relationship between the increase in testosterone levels and the hypothalamic system. The oral ingestion of coumestrol-rich bean sprouts increases serum testosterone, which can also suppress the progression of aging.

## 5. Conclusions

It has been reported that HASPIN inhibitors suppress the proliferation of various cancer cells. In this study, the ingestion of bean sprouts containing large amounts of the HASPIN inhibitor, coumestrol, suppressed the progression of intestinal polyp development, cachexia, and hypogonadism, and increased serum testosterone in mice. Analysis of the detailed molecular mechanism responsible for these results is important for its application to humans.

## Figures and Tables

**Figure 1 biology-13-00736-f001:**
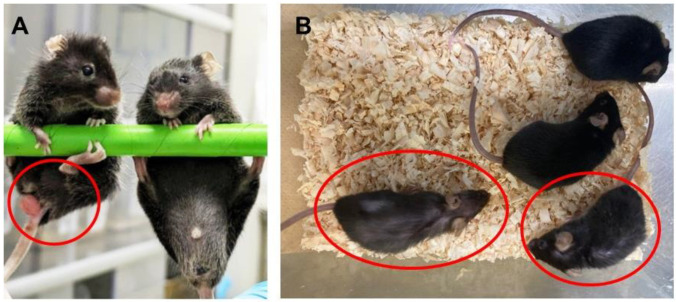
Appearance of mice fed the diet with bean sprouts. (**A**) Colon cancer was observed in *Apc^Min/+^* mice fed the standard diet (red circles on the left), but no colon cancer was observed in mice fed the diet with bean sprouts (right). (**B**) Hair loss was observed in aged wild-type mice (red circles) fed the standard diet, but not in 1-year-old littermates fed the diet with bean sprouts.

**Figure 2 biology-13-00736-f002:**
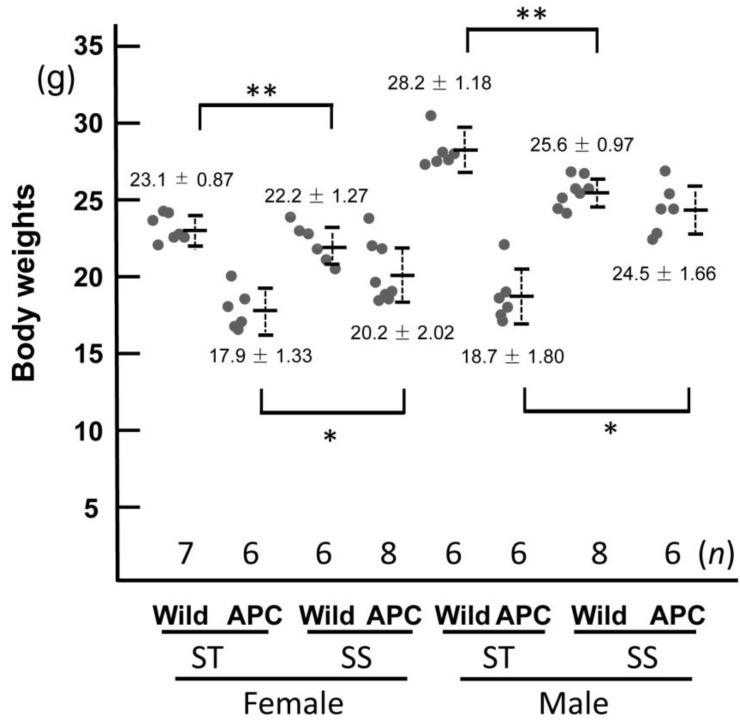
Body weights of mice fed the diet with bean sprouts. Male and female mice were fed a standard diet (ST) or the diet with bean sprouts (SS) from 7 to 12 weeks of age. Wild and APC indicate wild-type and *Apc^Min/+^* mice, respectively. Dots indicate individual values for each mouse, numbers indicate the average and standard deviation (±) for each group, and *n* indicates population size. Body weights in both sexes in both groups were significantly reduced by the diet with bean sprouts. It was significant differences (*p* < 0.05) between the same asterisks number (*, **).

**Figure 3 biology-13-00736-f003:**
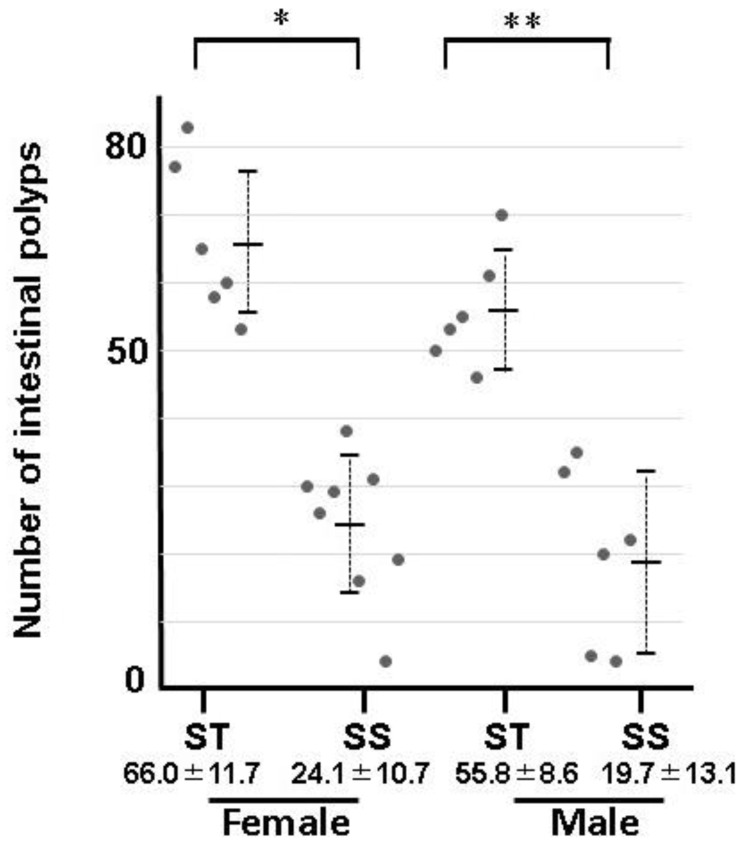
Numbers of intestinal polyps in *Apc^min/+^* mice fed the diet with bean sprouts. Male and female mice were fed a standard diet (ST) or the diet with bean sprouts (SS) from 7 to 12 weeks of age. Dots indicate results in female or male mice fed the standard diet or the diet with bean sprouts. Numbers indicate the average and standard deviation for each group. The development of polyps in females (*) and males (**) was significantly suppressed by the diet with bean sprouts (*p* < 0.05). *n* ≥ 6.

**Figure 4 biology-13-00736-f004:**
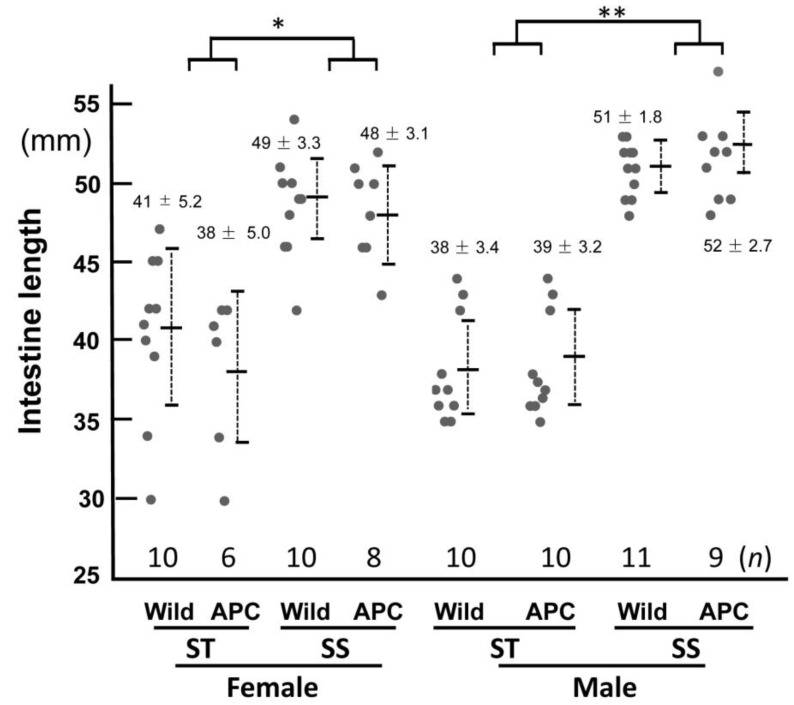
Intestinal lengths of Apc^Min/+^ mice fed the diet with bean sprouts. The intestines of mice fed the normal diet (ST) or a diet containing bean sprouts (SS) are shown. Wild and APC indicate wild-type and *Apc^Min/+^* mice, respectively. Dots indicate individual values for each mouse, numbers indicate the average and standard deviation for each group, and N indicates the population size. Intestinal length of polyps in females (*) and males (**) was significantly suppressed by the diet with bean sprouts (*p* < 0.05). *n* ≥ 6.

**Figure 5 biology-13-00736-f005:**
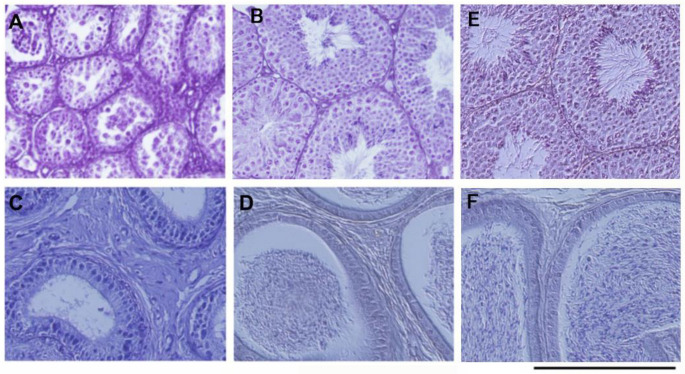
Hematoxylin-eosin staining of Bouin’s solution-fixed testes and epididymides sections from *Apc^Min/+^* mice. Spermatogenesis did not occur in the testis (**A**) or the epididymis (**C**) of *Apc^Min/+^* mice fed the standard diet. Spermatogenesis fully recovered in the testis (**B**) and the epididymis (**D**) of *Apc^Min/+^* mice fed the diet with bean sprouts compared to wild-type mice ((**E**) and (**F**), respectively). Bar = 200 μm.

**Figure 6 biology-13-00736-f006:**
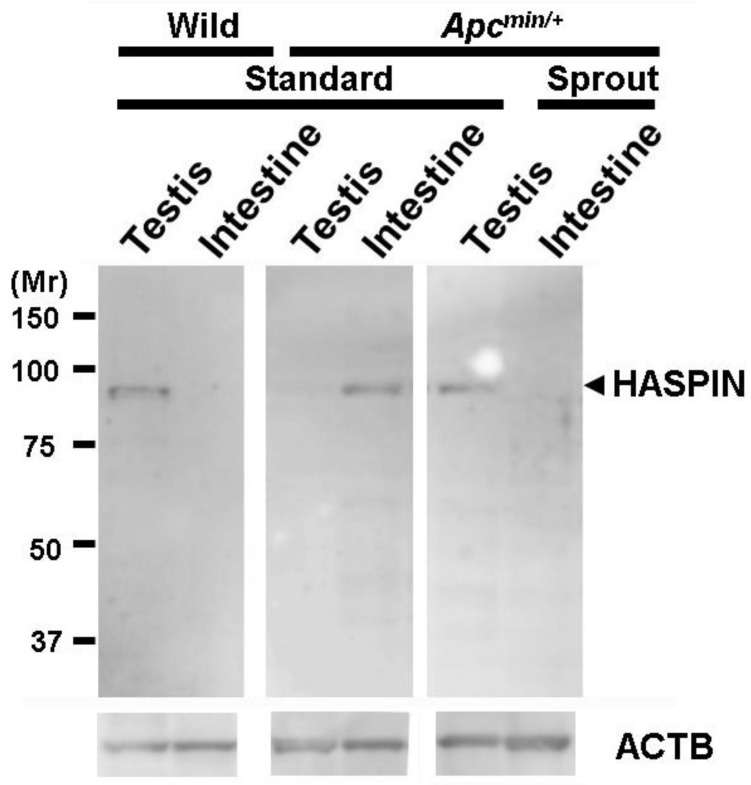
Identification of HASPIN expression in the testes and intestine using anti-HASPIN antibody. HASPIN was specifically expressed in the testes during spermatogenesis, but this expression was not observed in the testes of *Apc^Min/+^* mice fed the standard diet. It was expressed in the intestine where polyps developed in *Apc^Min/+^* mice fed the standard diet. It was also expressed in the testes of *Apc^Min/+^* mice fed the diet with bean sprouts, but no expression was observed in the intestine of *Apc^Min/+^* mice fed the diet with bean sprouts. The uncropped western blot figures were presented in Appendix A.

**Table 1 biology-13-00736-t001:** Testicular and epididymal weights of male mice fed the standard diet or diet with bean sprouts.

Bait Type	Standard		Sprout	
Genotype	WT	*Apc^Min/+^*	WT	*Apc^Min/+^*
Testicular weight (g)	0.09 ± 0.01	0.04 ± 0.03 *	0.09 ± 0.01	0.08 ± 0.02 *
Epididymal weight (g)	0.012 ± 0.0007	0.006 ± 0.0014 **	0.012 ± 0.0007	0.010 ± 0.0011 **
Epididymal fat weight (g)	0.3 ± 0.22 ***	0.0 ± 0.01 ****	0.21 ± 0.05 ***	0.21 ± 0.06 ****

Data are means ± standard deviation (*n* = 6). It was significant differences (*p* < 0.01) between the same asterisks number (*, **, ***, ****). ± means standard deviation. WT, wild type.

**Table 2 biology-13-00736-t002:** Serum testosterone levels in male mice fed standard or bean sprout-containing bait.

Bait Type	Standard		Sprout	
Genotype	WT	*Apc^Min/+^*	WT	*Apc^Min/+^*
Serum testosterone (pg/mL)	283.0 ± 51.0 *^,^ **	17.0 ± 6.6 *	2780.4 ± 1971.7 **	973.3 ± 785.7 **

Data are means ± standard deviation (*n* = 6). It was significant differences (*p* < 0.01) between the same asterisks number (*^,^ **). ± means standard deviation. WT, wild type.

## Data Availability

The datasets used and analyzed during this study are available from the corresponding author on reasonable request.

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
