# Peer review of "The Natural HASPIN Inhibitor Coumestrol Suppresses Intestinal Polyp Development, Cachexia, and Hypogonadism in a Mouse Model of Familial Adenomatous Polyposis (ApcMin/+)"

_biology, 2024, doi:10.3390/biology13090736_

Round 1

Reviewer 1 Report

Comments and Suggestions for Authors

Review Tanaka et al.,

In this manuscript, the authors provided some evidence that bean sprouts food supplementation reduced colorectal cancer progression and recued hypogonadism using the well-knowed colorectal adenomas APC knock out mouse model. The authors suggested that Coumestrol, a Haspin kinase inhibitor from bean sprouts extract, is linked to the observed phenotype. The authors demonstrated that bean sprouts food supplementation reduced body weight loss, epididymal fat loss and the number of polyps into the mouse’s intestines. They also demonstrated that bean sprouts food supplementation rescued partial spermatogenesis associated with a lack of expression of Haspin. Nevertheless, the manuscript lacks important information to conclude of the direct role of Coumestrol via the Haspin inhibition in the phenotype observed. Moreover, the way the authors address the anti-cancer/cachexia effect should be better characterized. Finally, the discussion should be more detailed.

Globally, the data are quite convincing but there are some points to consider before that the manuscript may be considered for publication in Biology.

Major comments:

1-   The authors did not fully demonstrate cachexia phenotype in this study. The author only associated the body weight loss and epididymal fat loss to cachexia, but cachexia is characterized strongly associated to muscle loss. Sampling of targeted organs including muscles such as gastrocnemius muscle and heart, and some additional fat tissues may be valuable for this study and will help to conclude. The pictures of Figure 1 cannot be used to identify cachexia.

2-   The Figure 1A are not convincing to justify colon cancer occurrence in APC-/- versus absence in the bean sprouts supplemented group.

3-   The number of mice per group is quite low and limit the conclusion of the study (n=6 per group?).

4-   The method to quantify intestinal polyps is not well described (staining, microscopy?). If pictures are available, they should be provided.

5-   The authors did not demonstrate that HASPIN inhibition is linked to cancer progression. No functional was used to characterize this effect. The use of Haspin kinase inhibitor might be used or a lack of function study may be proposed  to confirm the functional link. These data would be highly valuable for the interest of the manuscript.

6-   Similarly, Coumestrol extraction from sprout bean and direct pharmacological evaluation of an enriched fraction would confirm the author assumptions. This would be valuable for the manuscript since the authors confirmed that the ingestion of more than 1.5 kg of bean sprouts is not possible.

7-   The evaluation of functional spermatogenesis by histology is poorly convincing (Figure 2). Even if spermatogenesis deficient is described in the scientific literature, images of staining from wt mice would be valuable. Even if the table 2 demonstrated a reduction of testicular testis weight, images of testis would be valuable. Conversely, the analysis of serum testosterone is appreciated and confirmed the statement of the authors regarding partial spermatogenesis rescue upon bean sprout food supplementation.

8-   Statistical analysis should be re-evaluated to confirm normality and parametric assumptions of the data before choosing parametric test (see Gosselin, Laboratory Animals 2019, Vol. 53(1) 28–42 / Pollard et al., Molecular Biology of the Cell 2019, Vol. 30(12)). Indeed, the number of mice per group might be quite low (n=6?) to confirm the parametric assumption. Moreover, Unpaired t-test is not a relevant statistical evaluation to analyze group-dependent response due to bias in power analysis. One/Two way ANOVA or corresponding non-parametric test should be used. Globally, the precise number of mice per group should be mentioned in the figure legend.

9-   All the original full blots used for the protein expression quantification must be provided as supplementary data.

Minor comments:

1-       The manuscript title should be reformulated to attenuate the message in accordance with the current data.

2-       The authors should homogenize the asterisks in the tables, indeed there are between one to five stars but only P<0.05 is described in the legend.

3-       Time-dependent analysis of mouse body weight and not just endpoint should be provided since body weight loss is a long-lasting effect.

4-       Pictures of intestines, testis, and colons should be provided.

5-       The authors poorly described the state of art of Coumestrol/Haspin inhibition role in cancer. Moreover, how Coumestrol affect tumor signaling pathway in APC mice and activity should be more discussed. Additional work in the discussion will help to emphasize the interest of this research for the scientific community.

Author Response

To Reviewer 1

Thank you for finding our research worth and the valuable suggestions.

We added and changed sentences as reviewers’ suggestion.

Finally, I asked a professional editor to proofread our English.

Reviewer`s suggestion 1,2

1-   The authors did not fully demonstrate cachexia phenotype in this study. The author only associated the body weight loss and epididymal fat loss to cachexia, but cachexia is characterized strongly associated to muscle loss. Sampling of targeted organs including muscles such as gastrocnemius muscle and heart, and some additional fat tissues may be valuable for this study and will help to conclude. The pictures of Figure 1 cannot be used to identify cachexia.

Reviewer`s suggestion 2

2-   The Figure 1A are not convincing to justify colon cancer occurrence in APC-/- versus absence in the bean sprouts supplemented group.

Response to the reviewer

Thank you for your important suggestions. As the reviewer`s suggestion, we added new figure 2 and 3 and referred them and table S1 and added reference [26] in (lines 157 to 168) in 3.1 Body weight.

ApcMin/+ mice showed normal growth to adulthood similar to wild-type controls, but lost body weight and became infertile with the development of colon cancer [26] (Figure 1). We investigated the effects of ingestion of bean sprouts on the appearance of the mice. In the treatment group, Mice were fed the diet with bean sprouts between 7 and 12 weeks of age. ApcMin/+ mice developed cachexia during the period of treatment [26] (Figures 1 and 2, Supplementary Table S1). We measured the body weight of mice at the end of the experiment. The results showed significant recovery of body weight in mice fed the diet with bean sprouts compared with controls (Figures 2, and Table S1). Both body weight and epididymal fat content were lower in wild-type and ApcMin/+ mice that ingested a diet containing bean sprouts than in those given the standard diet (Table 1). Mice fed the diet containing bean sprouts continued to grow and exhibited minimal cachexia after 12 weeks (Figures 1, 2, and Supplementary Table S1).

Reviewer`s suggestion 3

3-   The number of mice per group is quite low and limit the conclusion of the study (n=6 per group?).

Response to the reviewer

Thank you for your important suggestions.

We have described N>5 in our manuscript in all tables by mistake, but they mean N≧5.

As reviewers’ suggestion, we added additional observations and showed results of N≧6 in revised manuscript (new figures and tables).

Reviewer`s suggestion 4

4-   The method to quantify intestinal polyps is not well described (staining, microscopy?). If pictures are available, they should be provided.

Response to the reviewer

As a reviewer’s suggestion, we added the sentences and the reference in 2.3 Intestinal polyp counts

(lines 116 to122) and added reference [30]. And we added supplemental figure S2 and S3

Intestinal polyps were counted using the published method [29]. The length of freshly removed intestine was measured, and each small intestine was divided into five equal segments. The segments were incised longitudinally, washed with phosphate-buffered saline, laid flat on filter paper, and fixed for 24 h with 10% neutral-buffered formalin. The fixed intestinal segments were stained with 1% methylene blue and examined for tumors by gross inspection and light microscopy. Polyps more than 2 mm in diameter were classified as precancerous colon polyps [30].

Reviewer`s suggestion 5

5-   The authors did not demonstrate that HASPIN inhibition is linked to cancer progression. No functional was used to characterize this effect. The use of Haspin kinase inhibitor might be used or a lack of function study may be proposed to confirm the functional link. These data would be highly valuable for the interest of the manuscript.

Response to the reviewer

I agree with the reviewer's suggestion. As with our previous method [33], we are currently analyzing the effects of intraperitoneally administering coumestrol to familial colon cancer model mice.

Regarding the relationship between coumestrol and HASPIN, it is difficult to assert the relationship as pointed out by the reviewer. Here, we have described it as

The APC protein is a tumor suppressor that regulates the Wnt signaling pathway through the formation of complexes with various other proteins, and plays a role in the early stages of colorectal cancer development. Wnt/β-catenin signaling is also known to play an important role in the differentiation and proliferation of Leydig cells in the testis [25]. As administration of CHR-6494 suppressed the development of polyps and hypogonadism in ApcMin/+ mice, HASPIN may not only function in chromosome segregation but also participate in the Wnt/β-catenin signaling pathway. Our results are identical to findings reported previously for the HASPIN inhibitor, CHR6494 [33]. Therefore, the HASPIN inhibitor coumestrol within bean sprouts is considered to have been the main factor involved in suppressing cancer. (lines 234 to 244 in discussion)

Reviewer`s suggestion 6

6-   Similarly, Coumestrol extraction from sprout bean and direct pharmacological evaluation of an enriched fraction would confirm the author assumptions. This would be valuable for the manuscript since the authors confirmed that the ingestion of more than 1.5 kg of bean sprouts is not possible.

Response to the reviewer

Thank for your suggestion.

Our recent research has demonstrated that even small amounts of bean sprouts may have these effects.  As based on these results, we are planning clinical trials in humans.

Reviewer`s suggestion 7

7-   The evaluation of functional spermatogenesis by histology is poorly convincing (Figure 2). Even if spermatogenesis deficient is described in the scientific literature, images of staining from wt mice would be valuable. Even if the table 2 demonstrated a reduction of testicular testis weight, images of testis would be valuable. Conversely, the analysis of serum testosterone is appreciated and confirmed the statement of the authors regarding partial spermatogenesis rescue upon bean sprout food supplementation.

Response to the reviewer

As a reviewer’s suggestion, we added the figure of wild type testis in new figure 5.

Reviewer`s suggestion 8

8-   Statistical analysis should be re-evaluated to confirm normality and parametric assumptions of the data before choosing parametric test (see Gosselin, Laboratory Animals 2019, Vol. 53(1) 28–42 / Pollard et al., Molecular Biology of the Cell 2019, Vol. 30(12)). Indeed, the number of mice per group might be quite low (n=6?) to confirm the parametric assumption. Moreover, Unpaired t-test is not a relevant statistical evaluation to analyze group-dependent response due to bias in power analysis. One/Two way ANOVA or corresponding non-parametric test should be used. Globally, the precise number of mice per group should be mentioned in the figure legend.

Response to the reviewer

As a reviewer’s suggestion, we added the sentence in 2.7 Statistical analyses (lines152 to 153), and

new figure 2, 3.

Statistical analysis was performed using Student’s t-test and one-way analysis of

variance (ANOVA) with Dunnett’s or Tukey’s test (lines152 to 153).

Reviewer`s suggestion 9

9-   All the original full blots used for the protein expression quantification must be provided as supplementary data.

Response to the reviewer

As a reviewer’s suggestion, we changed new figure 6.

To examine whether HASPIN is expressed in proliferating polyps, we performed a Western blotting analysis using an antibody to HASPIN. As a result, no HASPIN signal was observed in the small intestine without polyps, but we observed a HASPIN signal in small intestinal polyps, which we report here. HASPIN protein is very unstable, and it is very difficult to observe a signal of normal size, although this may depend on the condition of the polyp and the area sampled. We have conducted this experiment many times, and have published the results that are easy to understand here, and will attach the results as an additional supplement. At present, quantitative analysis is not possible with our technology.

Please see, additional figure 1 to 3.

Minor comments:

Reviewer`s suggestion 1

1-       The manuscript title should be reformulated to attenuate the message in accordance with the current data.

Response to the reviewer

As a reviewers’ suggestions, we changed the title as follows

The Natural HASPIN Inhibitor Coumestrol Suppresses Intestinal Polyp Development, Cachexia, and Hypogonadism in a Mouse Model of Familial Adenomatous Polyposis (ApcMin/+)

Reviewer`s suggestion 2

2-       The authors should homogenize the asterisks in the tables, indeed there are between one to five stars but only P<0.05 is described in the legend.

Response to the reviewer

As a reviewer’s suggestion, we added the sentence “It was significant differences (N<0.05) between the same asterisks number.” in all tables.

Reviewer`s suggestion 3

3-       Time-dependent analysis of mouse body weight and not just endpoint should be provided since body weight loss is a long-lasting effect.

Response to the reviewer

We don't have the data of the reviewer’s suggestion. We would show that the cancer progression was ultimately suppressed after Ingestion of the Bean Sprouts. We would like get more details on that in the next investigation.

Reviewer`s suggestion 4

4-       Pictures of intestines, testis, and colons should be provided.

Response to the reviewer

As a reviewer’s suggestion, we added the figure S1, S2 and S3.

Reviewer`s suggestion 5

5-       The authors poorly described the state of art of Coumestrol/Haspin inhibition role in cancer. Moreover, how Coumestrol affect tumor signaling pathway in APC mice and activity should be more discussed. Additional work in the discussion will help to emphasize the interest of this research for the scientific community.

Response to the reviewer

As a reviewer’s suggestion, we added sentences in Discussion (lines 234 to 244)..

The APC protein is a tumor suppressor that regulates the Wnt signaling pathway through the formation of complexes with various other proteins, and plays a role in the early stages of colorectal cancer development. Wnt/β-catenin signaling is also known to play an important role in the differentiation and proliferation of Leydig cells in the testis [25]. As administration of CHR-6494 suppressed the development of polyps and hypogonadism in ApcMin/+ mice, HASPIN may not only function in chromosome segregation but also participate in the Wnt/β-catenin signaling pathway. Our results are identical to findings reported previously for the HASPIN inhibitor, CHR6494 [33]. Therefore, the HASPIN inhibitor coumestrol within bean sprouts is considered to have been the main factor involved in suppressing cancer.

Reviewer 2 Report

Comments and Suggestions for Authors

In this paper, the authors show some interesting insight on the potential benefit of a dietary-based delivery of Haspin inhibitors to counteract/prevent the insurgence of colorectal cancer. Overall, the conclusions drawn by the authors are quite well-supported by the data, and might be of interest to a relatively wide audience.

However, I felt a bit disappointed regarding the introduction and discussion session, in my opinion a lot of additional notions are to be provided to the reader for a more exhaustive depiction of the genetic background of the mice, the kinase Haspin and how this work might integrate with recent reports in the Haspin field.

I think this work would be improved if the authors could consider the following points:

- Overall the whole point of the paper is that, in murine CRC models, bean sprouts administration elicits a beneficial repressive effect on tumorigenesis through Haspin inhibition. Beside the western blot in figure 3 shows a change in the pattern of haspin expression upon bean sprouts administration, it does not demonstrate that Haspin is effectively inhibited. The evaluation of H3-T3p would be an extremely meaningful addition to the panel.

- No detailed information of the murine model are provided. Beside APC being a well-known factor in CRC development, I suggest the authors to add a brief description of what APC is, why it can be used to trigger colorectal carcinogenesis, and how it does so in their introduction. For example, the paragraph about ApcMin/+ model present in the discussion could be moved to the introduction.

- In the abstract the authors state that haspin phosphorylates H3-T3 in
mitosis. This is not entirely true: while the level of phosphorylation
of H3-T3 peaks in mitosis, this PTM is present also in interphase
(Fresàn, Rodríguez-Sánchez et al 2020) and G0 (Quadri, Setric et al
2021). In particular, suggest the author to include these observations
in their introduction, also disserting the fact that even non-cycling
tissues and the functionality of their primary cilia are going to be
affected (Quadri, Sertic et al 2021).” 

- Considering that the authors are proposing the administration of Haspin inhibitors to be beneficial to prevent/treat colorectal cancer, I believe that all the known Haspin function potentially being abrogated in the cells by coumestrol administration should be considered or at least stated, rather than just saying "Haspin [...] plays various regulatory roles in cellular function". In particular, I would explicitly refer to works reporting Haspin roles potentially relevant to carcinogenesis, namely: CPC recruitment (Wang 2011; Yamagishi 2010; Kelly 2010); tissue morphogenesis and homeostasis (Gao 2023; Quadri 2023); Pds5 and cohesin regulation (Zhou 2017; Jiang 2018); polycomb regulation (Fresàn 2020). This might also be instrumental to the authors to elaborate possible side-effects of coumestrol administration.

- Par 3.4: HASPIN is exclusively expressed in haploid germ cells. Besides, it was originally identified in such cells, the expression levels of this protein were later reported to correlate to the proliferative state of the tissue (Nishida-Fukuda 2021). This would be in agreement with the expression pattern reported in figure 3 and fit with the authors' sentence " [...] indicating that HASPIN may be a useful cancer marker". 

- while the authors state control animals were fed a 5L37 LABDIET, I believe most of the audience might be unfamiliar with it, so I suggest authors to briefly describe its composition.

- I believe the work would benefit from the addition of a one sentence introduction to each result paragraph, to help the reader feel less "lost" when jumping to one phenotype to the other and guide him/her through a more enjoyable reading experience (eg, (besides being obvious), why did the authors look for intestinal polyps (par. 3.2) or testes defects (par 3.3)?)

- Legends report N>5. Reporting the exact number would be a nice addition.

Author Response

To Reviewer 2

Thank you for finding our research worth and the valuable suggestions.

Reviewer`s suggestion 1

- Overall the whole point of the paper is that, in murine CRC models, bean sprouts administration elicits a beneficial repressive effect on tumorigenesis through Haspin inhibition. Beside the western blot in figure 3 shows a change in the pattern of haspin expression upon bean sprouts administration, it does not demonstrate that Haspin is effectively inhibited. The evaluation of H3-T3p would be an extremely meaningful addition to the panel.

Response to the reviewer

We agree with reviewer’s suggestion.  As the reviewer’s suggestion, we try to analysis of Histone H3 and H3-T3p expressions by western blotting. However, we were unable to obtain stable analytical results using our technology. In the future, we would like to clarify the relationship between HASPIN and H3-T3p in cancer.

Reviewer`s suggestion 2

- No detailed information of the murine model are provided. Beside APC being a well-known factor in CRC development, I suggest the authors to add a brief description of what APC is, why it can be used to trigger colorectal carcinogenesis, and how it does so in their introduction. For example, the paragraph about ApcMin/+ model present in the discussion could be moved to the introduction.

Response to the reviewer

As the reviewer’s suggestion, we changed and added the sentences in the introduction (

lines 73 to 79).

Intraperitoneal administration of the artificial HASPIN inhibitor CHR-6494 was shown to suppress colon cancer in a mouse model of familial adenomatous polyposis (ApcMin/+). The ApcMin/+ mouse is a familial colon cancer mouse model with a nonsense mutation at codon 850 in the Apc gene [25]. APC protein forms a complex with various other proteins and is involved in the control of signal transduction pathways [25, 26]. It may be a tumor suppressor involved in the earliest stages of human colorectal cancer development.

Reviewer`s suggestion 3

- In the abstract the authors state that haspin phosphorylates H3-T3 in mitosis. This is not entirely true: while the level of phosphorylation of H3-T3 peaks in mitosis, this PTM is present also in interphase (Fresàn, Rodríguez-Sánchez et al 2020) and G0 (Quadri, Setric et al, 2021). In particular, suggest the author to include these observations in their introduction, also disserting the fact that even non-cycling tissues and the functionality of their primary cilia are going to be affected (Quadri, Sertic et al 2021).” 

Response to the reviewer

As the reviewer’s suggestion, we changed and added the sentences in the introduction (lines 58 to 60).

HASPIN phosphorylates histone H3 at Thr3 in mitotic cells. While the level of H3-T3 phosphorylation peaks during mitosis it also occurs during interphase [11] and G0 [12], and even non-cycling tissues and the function of their primary cilia are affected [12].

Reviewer`s suggestion 4

- Considering that the authors are proposing the administration of Haspin inhibitors to be beneficial to prevent/treat colorectal cancer, I believe that all the known Haspin function potentially being abrogated in the cells by coumestrol administration should be considered or at least stated, rather than just saying "Haspin [...] plays various regulatory roles in cellular function". In particular, I would explicitly refer to works reporting Haspin roles potentially relevant to carcinogenesis, namely: CPC recruitment (Wang 2011; Yamagishi 2010; Kelly 2010); tissue morphogenesis and homeostasis (Gao 2023; Quadri 2023); Pds5 and cohesin regulation (Zhou 2017; Jiang 2018); polycomb regulation (Fresàn 2020). This might also be instrumental to the authors to elaborate possible side-effects of coumestrol administration.

Response to the reviewer

As the reviewer’s suggestion, we changed and added the sentences in the introduction (

lines 66 to 68).

It also has functions that are relevant to carcinogenesis, including CPC recruitment [17–19], tissue morphogenesis and homeostasis [20, 21], Pds5 and cohesin regulation [22, 23], and polycomb regulation [11].

Reviewer`s suggestion 5

- Par 3.4: HASPIN is exclusively expressed in haploid germ cells. Besides, it was originally identified in such cells, the expression levels of this protein were later reported to correlate to the proliferative state of the tissue (Nishida-Fukuda 2021). This would be in agreement with the expression pattern reported in figure 3 and fit with the authors' sentence " [...] indicating that HASPIN may be a useful cancer marker". 

Response to the reviewer

As the reviewer’s suggestion, we added the sentence in 3.4 HASPIN expression. (lines 192 to 195).

HASPIN is exclusively expressed in haploid germ cells and not in undifferentiated testicular germ cells (Figure 6) [34]. The expression levels of this protein are correlated with the proliferative state of the tissue [35].

Reviewer`s suggestion 6

- while the authors state control animals were fed a 5L37 LABDIET, I believe most of the audience might be unfamiliar with it, so I suggest authors to briefly describe its composition.

Response to the reviewer

As a reviewer’s suggestion, we add the sentence in materials and methods (lines 99 to 100)

The mice were fed the specialized laboratory rodent diet 5L37 LABDIET (Japan SLC, Inc.), which is available internationally as a LabDiet® product (Land O’ Lakes, Inc., Arden Hills, MN, USA).

Reviewer`s suggestion 7

- I believe the work would benefit from the addition of a one sentence introduction to each result paragraph, to help the reader feel less "lost" when jumping to one phenotype to the other and guide him/her through a more enjoyable reading experience (eg, (besides being obvious), why did the authors look for intestinal polyps (par. 3.2) or testes defects (par 3.3)?)

Response to the reviewer

Thank you for your suggestion. As the reviewer’s suggestion, we added sentences on each paragraph in results (lines 157 to 159, lines 171 to 174, lines 182 to183, and lines 193 to195)

3.1 Body weight

ApcMin/+ mice showed normal growth to adulthood similar to wild-type controls, but lost body weight and became infertile with the development of colon cancer [26] (Figure 1).

3.2 Intestinal polyps

Intestinal polyps were observed in ApcMin/+ mice fed the diet with bean sprouts between 7 and 12 weeks of age. Intestinal polyps become larger with age in ApcMin/+ mice [30]. We investigated the effects of bean sprout ingestion on the number of polyps more than 2 mm in diameter, defined as precancerous colon polyps [30].

3.3 Observations of testes and epididymides

ApcMin/+ mice develop hypogonadism [31, 32]. The HASPN inhibitor, CHR-6494, suppresses the development of hypogonadism [33].

3.4 HASPIN expression

HASPIN is exclusively expressed in haploid germ cells and not in undifferentiated testicular germ cells (Figure 6) [34]. The expression levels of this protein are correlated with the proliferative state of the tissue [35]. Reviewer`s suggestion 8

Reviewer`s suggestion 8

- Legends report N>5. Reporting the exact number would be a nice addition.

Response to the reviewer

Thank you for your important suggestions.

We have described N>5 in our manuscript in all tables by mistake, but they mean N≧5.

As reviewers’ suggestion, we added additional observations and showed results of N≧6 in revised manuscript (new figures and tables).

Reviewer 3 Report

Comments and Suggestions for Authors

In this paper, the author executed mouse study using familial adenomatous polyposis mouse model, and concluded that the bean sprite diet has a beneficial effect on intestinal polyps, cachexia, and hypogonadism. This conclusion is supported by data on animal level (body weight, intestine length, appearance, etc), tissue level (Spermatogenesis), and molecular level (HSPIN expression). The beneficial effect from bean sprout indigestion with regard to colorectal cancer is strongly demonstrated in this study.

However, more evidence is needed to support the axis of Coumestrol-HASPIN in observed phenotype. To claim the role of Coumestrol-HASPIN is relevant, a few mechanistic studies should be executed: for example, in mouse models where HASPIN is suppressed, the beneficial effects of bean sprouts should be diminished; alternatively, the HASPIN kinase activities should be lower in mouse model fed with bean sprout. Without mechanistic studies, it is hard to claim the relevance of Coumestrol and HASPIN in obsessed phenotype.

Author Response

To Reviewer 3

Comments and Suggestions for Authors

In this paper, the author executed mouse study using familial adenomatous polyposis mouse model, and concluded that the bean sprite diet has a beneficial effect on intestinal polyps, cachexia, and hypogonadism. This conclusion is supported by data on animal level (body weight, intestine length, appearance, etc), tissue level (Spermatogenesis), and molecular level (HSPIN expression). The beneficial effect from bean sprout indigestion with regard to colorectal cancer is strongly demonstrated in this study.

However, more evidence is needed to support the axis of Coumestrol-HASPIN in observed phenotype. To claim the role of Coumestrol-HASPIN is relevant, a few mechanistic studies should be executed: for example, in mouse models where HASPIN is suppressed, the beneficial effects of bean sprouts should be diminished; alternatively, the HASPIN kinase activities should be lower in mouse model fed with bean sprout. Without mechanistic studies, it is hard to claim the relevance of Coumestrol and HASPIN in obsessed phenotype.

To Reviewer 3

Thank you for finding our research wealth and the valuable suggestions.

However, more evidence is needed to support the axis of Coumestrol-HASPIN in observed phenotype. To claim the role of Coumestrol-HASPIN is relevant, a few mechanistic studies should be executed: for example, in mouse models where HASPIN is suppressed, the beneficial effects of bean sprouts should be diminished; alternatively, the HASPIN kinase activities should be lower in mouse model fed with bean sprout. Without mechanistic studies, it is hard to claim the relevance of Coumestrol and HASPIN in obsessed phenotype.

I have produced and been analyzing Haspin knockout mice.

Since Haspin knockout mouse was not knocked out in a condition-dependent manner, it is difficult to analyze its direct role in cancer. Here, we compare the results with those obtained with a HASPIN-specific inhibitor and discuss the results. And As with our previous method [33] in references, we are currently analyzing the effects of intraperitoneally administering coumestrol to familial colon cancer model mice.

Regarding the relationship between coumestrol and HASPIN, it is difficult to assert the relationship as pointed out by the reviewer. Here, we have described it in discussion (lines 253 to 257) as follows.

“Our results may have been due to the effects of dietary fiber and ingestion of polyphenols, as well as the multiple pharmacological effects of coumestrol as a HASPIN inhibitor. Moreover, the anticancer effects of bean sprout ingestion may have been due to interactions between coumestrol and other nutrients, and/or metabolites produced by intestinal bacteria.

As with our previous method [34, Tanaka, H.; Wada, M.; Park, J. HASPIN kinase inhibitor CHR-6494 suppresses intestinal polyp development, cachexia, and hypogonadism in Apcmin/+ mice. Eur J Cancer Prev. 2020, 29, 481-485. https://doi.org/10.1097/CEJ.0000000000000562.], we are currently analyzing the effects of intraperitoneally administering coumestrol to familial colon cancer model mice.
